# Diet Quality Influences the Occurrence of Food Aversions in Women Undergoing Adjuvant Chemotherapy for Breast Cancer

**DOI:** 10.3390/ijerph192113915

**Published:** 2022-10-26

**Authors:** Luiza Kuhnen Reitz, Jaqueline Schroeder, Marina Raick, Patricia de Fragas Hinnig, Francilene Gracieli Kunradi Vieira, Maria Alice Altenburg De Assis, Edson Luiz Da Silva, Giuliano Di Pietro, Patricia Faria Di Pietro

**Affiliations:** 1Post-Graduation Program in Nutrition, Federal University of Santa Catarina, Florianópolis 88040900, Brazil; 2Graduation Program in Nutrition, Federal University of Santa Catarina, Florianópolis 88040900, Brazil; 3Clinical Analysis Department, Federal University of Santa Catarina, Florianópolis 88040900, Brazil; 4Pharmacy Department, Federal University of Sergipe, Aracaju 49100000, Brazil

**Keywords:** diet quality, antioxidant intake, food aversions occurrence, oxidation, chemotherapy, adjuvant treatment, breast cancer

## Abstract

Food aversions in women undergoing adjuvant chemotherapy for breast cancer may be linked to oxidative stress and gastrointestinal consequences underlying it, and diet possibly plays a role in this association. This follow-up study included 73 women with breast cancer treated in Florianopolis City, Brazil. Dietary antioxidant capacity–DaC (mmol/d), diet quality–Brazilian Healthy Eating Index Revised (BHEI-R score), and oxidative stress biomarkers were accessed before the treatment, and women were asked if they developed food aversions during adjuvant chemotherapy. Red meat was the main aversion-causing food reported (37.9%, n = 9). There was no difference in DaC, BHEI-R score, or oxidative stress biomarkers between women with no food aversion occurrence and those showing food aversions. A logistic regression adjusted model showed that women exhibiting higher BHEI-R scores were 1.08 times more likely to not develop food aversions during adjuvant chemotherapy (*p* = 0.041). In summary, this innovative investigation showed that diet quality before adjuvant chemotherapy may influence the non-occurrence of food aversion. Considering this, the result opens new areas for early nutritional interventions, focusing on reducing the occurrence of food aversions and consequently benefiting women with breast cancer by having better outcomes in oncologic treatment.

## 1. Introduction

Breast cancer is the neoplasm with the highest incidence in females worldwide, with 2.3 million new cases, according to estimates carried out in 2020 [1]. In Brazil, 66,280 new diagnoses of breast cancer are expected for each year of the 2020–2022 triennium [2].

Among the strategies for dealing with the disease, adjuvant treatment stands out, including chemotherapy, hormone therapy, and biological therapy [3]. However, it is known that chemotherapy is often accompanied by side effects such as nausea, emesis, appetence, early satiety, xerostomia, parasomnia, and dysgeusia. Such factors trigger food aversions that lead to lower calorie and protein intake and, consequently, expose patients to weight loss and greater risk of malnutrition [4]. Thus, the occurrence of food aversions might be harmful to breast cancer patients’ relationship with food intake, nutritional status [5], quality of life, and prognosis of the disease [6].

Although the association between adjuvant chemotherapy and food aversions is already well understood [7], gustatory and olfactory alterations, responsible for food aversions, are also caused by nutritional deficiencies, mainly zinc [8] and vitamin B_12_ [9]. In addition, one of the main causes of food aversions in response to adjuvant chemotherapy is increased oxidative stress [10]. We have already demonstrated that diet quality and a diet rich in antioxidants can help mitigate the excess production of reactive oxygen species [11,12]. 

However, to date, no studies were identified evaluating the influence of Dietary antioxidant Capacity (DaC), diet quality, and oxidative stress before adjuvant chemotherapy on the occurrence of food aversions in breast cancer patients. In this sense, we aimed to investigate the influence of DaC, diet quality, and oxidative stress, before the adjuvant treatment on the occurrence of food aversions in women with breast cancer. We hypothesize that a healthier diet, before the adjuvant treatment in women with breast cancer, characterized by higher antioxidant capacity and diet quality, contributes to decreasing the occurrence of food aversions undergoing adjuvant treatment.

## 2. Materials and Methods

### 2.1. Study Design and Sampling

This is a follow-up study with a sample of women diagnosed with breast cancer recruited between 2006 and 2011. The patients were hospitalized for breast tumor removal surgery in a public hospital reference in breast cancer surgical treatment and subsequently underwent adjuvant treatment in a referral hospital in oncologic treatment, in Florianopolis city, Southern Brazil.

Eligibility criteria consisted of suspicion of malignant breast tumor or diagnosis of tumor malignancy, which was confirmed by anatomopathological examination, and submission to adjuvant chemotherapy as part of the breast cancer treatment. Exclusion criteria included: previous history of neoplasia; surgical procedures in a time equal to or less than one year; pregnant or lactating women; confirmation of benign breast tumor without suspicion of malignancy; diagnosis of neurological disease; positive diagnosis for acquired immunodeficiency virus (HIV+); and women previously submitted to neoadjuvant cancer treatment.

Data were collected before the adjuvant chemotherapy, and the assessment of the occurrence of food aversion was performed after the end of all chemotherapy sessions. Baseline data collection occurred during the same period as the anatomopathological examination.

All participants received information about the study protocol, and only those who signed the informed consent term were included in the sample. This study was conducted according to ethical principles and under the Ethics Committees’ approval (protocol numbers: 0012.0.233.242-10, 015/2009, 099/2008, and 492/2009).

### 2.2. Dietary Assessment

Food intake data were collected using a quantitative food frequency questionnaire (FFQ), which includes 112 food items, validated for the Brazilian population [13] and adapted by Medeiros [14]. The FFQ information reflects the food consumption, which occurred in the year before the diagnosis.

### 2.3. Dietary Antioxidant Capacity Assessment

For DaC calculation purposes, it was used a database with 3139 foods described, based on FRAP—ferric reducing antioxidant power method of antioxidant capacity assessment. Those foods were from Scandinavia, the USA, Europe, South America, Africa, and Asia [15]. The FRAP from foods in the database was described in mmol/100 g, and for calculation was transformed in mmol/g. For some foods without information on antioxidant capacity, the data from other food with similar nutritional content in vitamins, minerals, and bioactive compounds with an antioxidant capacity [16,17,18,19,20] or from the same botanical group were used. When the information on the cooking method was not described on FFQ, the average raw and cooked value was used. For foods with the same characteristics described in FFQ, but with more than one option described in the database, the average of those options was used. DaC was expressed as the sum of antioxidant capacity from all foods ingested in a habitual diet (mmol/day). The antioxidant capacity (aC) from foods groups, was also evaluated (mmol/day), using the Food Guide for Brazilian Population [21] as reference for the division into groups: whole cereals, legumes, tubers and roots, total fruits, and total vegetables; additionally, considering the main antioxidant present in food, they were also grouped as follows: cruciferous vegetables, rich in isothiocyanates [22]; orange and dark green vegetables and fruits, rich in beta-carotene [21]; citric foods, rich in Vitamin C [21]; red vegetables and fruits, rich in lycopene [23]; and polyphenol-rich foods and beverages [24,25]. 

### 2.4. Diet Quality Assessment

Diet quality evaluation was performed by the Brazilian Healthy Eating Index Revised (BHEI-R) application [26], validated by Andrade et al. [27], which is based on dietary recommendations for the Brazilian population [28].

The FFQ data were transformed into quantitative nutrient information [29] using Microsoft Excel16.0^®^ software, based on the Brazilian Food Composition Database [21] and the Food Composition Database of the United States Department of Agriculture [22]. After this data transformation, the quantitative data were grouped into 12 components to evaluate diet quality through BHEI-R [26,27]: total fruits (including fruits and natural fruit juices); whole fruits; total vegetables; dark green and orange vegetables, and legumes; total grains (including grains, roots, and tubers); whole grains; milk and dairy products (including soy drinks); meat, eggs and legumes; oils (including mono and polyunsaturated fats, nuts, seeds, and fish oils); saturated fats; sodium; and calories from solid fats, alcohol and added sugars—SoFAAS. Legumes were included in components “total vegetables”, and “dark green and orange vegetables and legumes” only after reaching the maximum score for “meat, eggs and legumes” [26]. The serving sizes and/or quantity of each BHEI-R index component were expressed by energy density (number of servings/1000 kcal or mg of sodium/1000 kcal). The saturated fat and SoFASS intake were expressed as a proportion of total energy consumed [26].

The BHEI-R total score consists of the sum of scores provided by the 12 components, which can range from 0 to 100 points; the highest score is achieved when the diet agrees with key recommendations for a healthy diet, designed for the Brazilian population [28]. The score from the following components ranges from 0 to 5 points: total fruits (including fruits and natural fruit juices), whole fruits, total vegetables, dark green and orange vegetables and legumes, total grains (including grains, roots, and tubers), and whole grains; for the components milk and dairy products, meat, eggs and legumes, and oils, the score range from 0 to 10 points. The absence of consumption of all those components cited above leads to scoring zero, and the intermediate intake is proportionally calculated. The score for the components: saturated fat and sodium ranges from 0 to 10 and is inversely proportional to the intake; the highest and lowest score is defined as 7% and ≥15% of total energy expenditure (TEE) for saturated fat, and ≤0.7 g/1000 kcal and ≥2.0 g/1000 kcal for sodium, respectively. The score from SoFAAS varies between 0 and 20, and it is inversely associated with the intake, ranging from ≤10% (20 points) and ≥35% (0 points) of the TEE.

### 2.5. Food Aversions Assessment

Data on food aversions were identified through the Clinical-Nutritional Questionnaire applied after the end of the treatment [30]. The questionnaire includes the question: “Did you develop some kind of food aversion during treatment?”; the answer was dichotomized as “yes” or “no”, and when the participant answered yes, there was a blank space to include by writing which food aversion was developed. All participants were informed by the trained interviewer about the definition of food aversion resulting from the treatment, which here is understood as food that was previously normally consumed and became poorly tolerated, not tolerated, and unpalatable after the beginning of adjuvant chemotherapy.

### 2.6. Oxidative Stress Biomarkers Analysis

For oxidative stress biomarkers analysis, 15 mL of blood was collected through venipuncture in the forearm, on the diagnosis day in tubes with and without EDTA to obtain plasma and serum, respectively, proceeded by centrifugation (1000× *g*/10 min). 

An aliquot of whole blood was separated immediately after the blood collection, to perform the reduced glutathione (GSH) assay, using 20% trichloroacetic acid, as described by Beutler et al. [31]. FRAP method was applied in the determination of serum antioxidant capacity, based on Benzie and Strain [32]. The determination of lipid peroxidation was performed by the measurement of thiobarbituric reactive acid substances (TBARS) purposed by Esterbauer et al. [33], and lipid hydroperoxide (HL), following the methodology of Nourooz-Zadeh [34]. To evaluate the concentrations of carbonylated proteins (CP), a plasma protein oxidation assay described by Levine et al. [35] was performed. The LH, TBARS, and FRAP assays were executed on the blood collection day. For CP determination, an aliquot of plasma was stored at −80 °C for 30 days. The biomarkers analysis was performed in duplicate.

### 2.7. Other Parameters

A sociodemographic, clinical, and anthropometric questionnaire [30] was applied at the beginning of the study by a trained interviewer. This instrument contains questions about the identification of the patient, clinical and reproductive history, sociodemographic data, lifestyle (physical activity practice, smoking, and alcohol intake), family history of cancer, and anthropometric data [36].

Weight, height, and waist circumference (WC) data were collected according to standard methods by a trained researcher [37]. Height and weight were measured on a mechanical scale (Filizola ^®^ São Paulo, SP, Brazil) with a capacity of 150 kg and 100 g precision. The body mass index (BMI) was calculated by weight (kg) divided by height (m) squared and was classified according to the World Health Organization’s [38] cut-off points [39]. WC was collected using an inelastic tape and was classified using the WHO [39] cut-off classification.

### 2.8. Statistical Analysis

Data was entered on Microsoft Office Excel^®^ (2016) and analyzed on STATA^®^ 13.0 (STATA CORPORATION, 2009). The descriptive patient information on sociodemographic, clinical, and anthropometric data was presented by absolute and relative frequencies. The normality of continuous data was verified by skewness and kurtosis tests. Symmetric data were expressed as mean and standard deviation and the asymmetric data as median and interquartile range. 

The differences between women with and without food aversion occurrence for data of sociodemographic, clinical, anthropometric, and nutritional variables were evaluated by the Student *t*-test for Gaussian-distributed data and the Mann-Whitney test for non-parametric data. Pearson’s Chi-square test was used for evaluating the differences in categorical data, between women with and with no food aversion occurrence. DaC was adjusted for energy intake by the residual method [40]. Logistic regression models were used to evaluate the association between food aversion occurrence and the DaC, BHEI-R score, and oxidative stress biomarkers obtained before the beginning of adjuvant chemotherapy. In these models, the DaC, BHEI-R score, and oxidative stress were considered as the independent variables, while the non-occurrence of food aversions was the outcome. The variables evaluated in the present study with *p* ≤ 0.2 values in the crude logistic regression models were included in the adjusted model. Therefore, the logistic regression was adjusted by age, schooling, number of chemotherapeutic sessions, tumor stage, tumor size, and alcohol intake. The accepted statistical significance level was *p* < 0.05.

## 3. Results

The present study included 73 women with a mean age of 51.9 ± 11.6 years. With respect to participants’ features, 63% (n = 46) were overweight or obese, 91.8% (n = 67) were diagnosed with invasive carcinoma, and 32.9% (n = 24) developed food aversion during adjuvant chemotherapy. The average treatment duration was 14.1 ± 4.4 months during the time of collecting information on food aversion. Table 1 and Table 2 show that there were no differences in sociodemographic, clinical, anthropometric, nutritional features, and oxidative stress biomarkers between women who developed and not developed food aversion during treatment, except for tumor size, which was higher in women reporting food aversion occurrence indicating a more aggressive disease (*p* = 0.004). Although not significant, the tumor stages II and III (severity of the disease) were also more common in women with food aversion (75%) than in women with non-occurrence of food aversion (56%) (Table 1). 

Based on the BHEI-R, both groups of women had comparable total and stratified by components scores. Furthermore, data on the dietary antioxidant capacity and the antioxidant capacity of food groups also confirmed the similarity of the habitual diet of women with or without the occurrence of food aversion (Table 2).

The main reported foods regarding the occurrence of aversion are shown in Figure 1. It is noteworthy that most women (n = 17/24, 70.8%) had food aversion to two or more foods. Red meat was the main aversion-causing food reported by the women (37.9%, n = 9). Among the foods categorized as “other foods” (Figure 1), bananas, fish, eggs, fried snacks, and sweets, in general, stand out.

Multivariate logistic regression (Table 3) showed that women exhibiting a BHEI-R score before the beginning of the adjuvant treatment for breast cancer were 1.08 times (CI95% 1.003–1.154) more likely to not develop food aversions during the treatment (*p* = 0.041).

## 4. Discussion

This is an innovative investigation regarding the impact of dietary habits and oxidative stress biomarkers on the chance of developing food aversions in women undergoing adjuvant chemotherapy for breast cancer treatment. Women with breast cancer who exhibit higher diet quality scores before adjuvant treatment were more likely to not develop food aversions in response to chemotherapy, which confirmed our primary hypothesis. 

In a previous study from the same original sample, we have shown that a higher diet quality score was associated with higher ferric antioxidant power (FRAP), a biomarker of antioxidant defense in serum, before adjuvant treatment for breast cancer began [11]. Herein, the protective effect of diet quality before the beginning of adjuvant chemotherapy on the chance of developing food aversion may be linked to increased serum antioxidant defense. It is also possible that better diet quality before adjuvant treatment influenced the maintenance of a healthy diet during adjuvant chemotherapy, contributing to a higher intake of antioxidants during this period, which may attenuate oxidation [41] and consequently reduce gastrointestinal side effects, resulting in non-occurrence of food aversion [10,42]. 

On the contrary, DaC before adjuvant treatment was not associated with food aversion occurrence. This may be explained by the use of FFQ since it is a restricted list of foods and may not consider the diversity of items possibly consumed by these women [18]. Additionally, the BHEI-R considers the dietary balance between food groups, according to the recommendation for a healthy diet for the Brazilian population [21,26], which may be linked to other factors such as the inflammatory content of it [43] not taken into account by the DaC method [15]. Diet quality is related to the ingestion of food components such as fruits and vegetables that provides bioactive compounds, which can downregulate the transcription of pro-inflammatory cytokines, lowering systemic inflammation [44]. Those components (fruits and vegetables) are also related to fiber ingestion, which plays a role in the control of glycemic load and modulation of microbiota, contributing to reduced inflammation [45]. Finally, the intake of saturated fat, considered a component of BHEI-R, is related to unbalanced microbiota, increased intestinal permeability, and inflammation [46]. Moreover, oxidative stress biomarkers were not different between groups of women who developed and not developed food aversion. Orchard et al. [47] observed that women who recently finished the primary treatment for breast cancer, exhibiting higher diet quality scores by Healthy Eating Index 2010, showed lower levels of pro-inflammatory cytokines (interleukin-6, interleukin-17, and tumor necrosis factor-α receptor 2). Therefore, higher diet quality may be related to attenuation of the inflammatory process underlying the disease and adjuvant chemotherapy, leading to the reduction of toxicity and gastrointestinal side effects [48], finally influencing the occurrence of food aversion. In addition, inflammation may also play a role and be a consequence of higher tumor size [49,50] observed in women who reported food aversion, since it is a hallmark of breast cancer staging and aggressiveness [51]. 

Concerning the food aversions reported in the present investigation, meat was the primary, followed by soda and legumes. These findings are in accordance with the literature. Bours et al. [52] showed that colorectal cancer patients reported a decrease in the intake of meat, as the prevalent food was excluded from the diet. In a cohort with 696 cancer patients, Fassier et al. [53] found a significant reduction in meat intake following the cancer diagnosis, which was linked to lower protein intake compared to the period before the diagnosis. Previous investigations reported that meat is one of the main foods that women with breast cancer undergoing adjuvant chemotherapy have aversion to [7,54,55]. Herein, the aversion to meat may be harmful to women with breast cancer undergoing adjuvant chemotherapy, since it may lead to lower protein intake. Low consumption of protein-rich foods contributes to muscle loss, which is a burden of disease and adjuvant treatment due to pro-inflammatory and catabolic features, despite the increasing prevalence of overweight among breast cancer patients [56]. Muscle loss in cancer patients occurs in varying degrees and leads to worsened clinical outcomes including higher toxicity and reduced tolerance of treatment, increasing mortality [57,58]. Aleixo et al. [59] showed in a literature review that women with breast cancer had a 68% greater mortality risk and more chemotherapy toxicity severity compared to non-sarcopenic patients with breast cancer. To prevent muscle degradation, breast cancer patients should be encouraged to consume adequate animal-derived proteins, preferably, since they are superior sources of the anabolic effect compared to vegetable sources, and should represent the majority (60%) of protein sources in diet during cancer treatment [56].

This study has some limitations; the heterogeneity of treatment duration among the participants may have influenced the food aversion occurrence reported since it is linked to oxidation and inflammation that underlie these features. However, the adjuvant treatment’s duration was tested in logistic regression as a confounding factor, reducing the bias. Moreover, the FFQ may not reflect antioxidant intake at all, considering that women may have the habit of eating some food items not included in this closed list. The DaC’s database may not reflect the antioxidant intake, since there was no information on Brazilian sources for some foods. Another difficulty was the absence of some foods described in QFA in the database. 

Otherwise, this is the first publication regarding the impact of dietary habits before the beginning of adjuvant chemotherapy on food aversions occurrence. The present findings may be important in bringing scientific evidence for the development of nutritional guidelines as a device for clinical practice in breast cancer care for women at the moment of diagnosis, focusing on improving outcomes of treatment. Finally, the recognition of food aversions allows the development of tools for nutritional adequacy, considering that the exclusion of some food groups from the diet might lead to worsened clinical outcomes.

## 5. Conclusions

This innovative investigation showed that diet quality before adjuvant chemotherapy may influence the non-occurrence of food aversion, which opens new areas for early nutritional interventions in breast cancer, given the burden of the inadequate nutrient intake promoted by diverse factors in these patients. In addition, these findings may be important in the public health area by contributing to the development of nutritional guidelines for breast cancer women before adjuvant chemotherapy, allowing them to have better outcomes.

## Figures and Tables

**Figure 1 ijerph-19-13915-f001:**
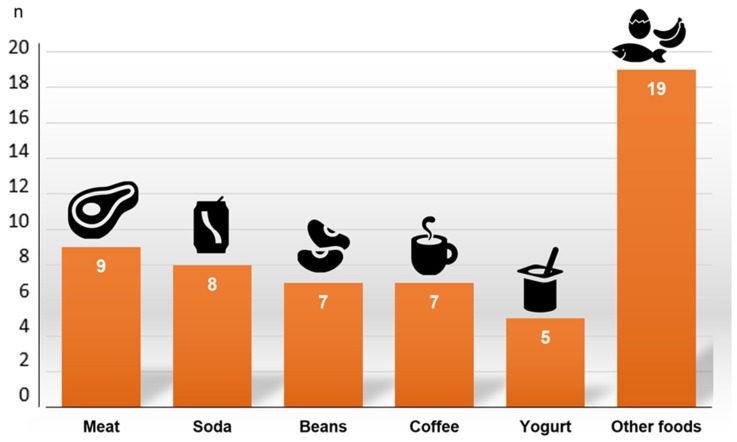
Aversion-causing foods reported by breast cancer patients undergoing adjuvant chemotherapy.

**Table 1 ijerph-19-13915-t001:** Sociodemographic, clinical, anthropometric, and nutritional characteristics of women with breast cancer, according to food aversion occurrence during adjuvant chemotherapy (n = 73).

	Occurrence of Food Aversion (n = 24)	Non-Occurrence of Food Aversion (n = 49)	*p*
Age (years) ^b^	50.6 (10.2)	52.5 (12.3)	0.518 ^$^
Ethnic group, n (%)			0.453 ^#^
White	22 (91.7)	47 (95.9)	
Non-white	2 (8.3)	2 (4.1)	
Schooling	4 (2, 8)	7 (4, 11)	0.104 ^&^
(years of study) ^a^			
Smoking, n (%)			0.875 ^#^
Yes	5 (20.8)	11 (22.4)	
No	19 (79.2)	38 (77.6)	
Alcohol intake, n (%)			0.725 ^#^
Yes	2 (8.3)	3 (6.10)	
No	22 (91.7)	46 (93.9)	
BMI (kg/m^2^) ^a^	26.3 (24.2, 31.8)	27.3 (23.7, 30.4)	0.809 ^&^
Waist circumference (cm) ^a^	87.5 (80.5, 101)	84 (78.5, 98)	0.537 ^&^
Tumor stage, n (%)			0.072 ^#^
0–I	6 (25)	23 (46.9)	
II–III	18 (75)	26 (56.1)	
Number of chemotherapy sessions	6 (6, 8)	6 (6, 8)	0.430 ^&^
Number of radiotherapy sessions	28.5 (28, 32)	32 (28, 33)	0.187 ^&^
Hormone therapy, n (%)			0.261 ^#^
Yes	18 (75)	42 (85.7)	
No	6 (25)	7 (14.3)	
Surgery type, n (%)			0.360 ^#^
Partial mastectomy	10 (41.7)	26 (53.1)	
Total mastectomy	14 (58.3)	23 (46.9)	
Tumor size (cm) ^a^	2.4 (1.8, 3.5)	1.5 (1, 2.5)	**0.004** ^&^

BMI: body mass index; BHEI-R: Brazilian Healthy Eating Index Revised; DaC: dietary antioxidant capacity; ^a^ median and interquartile range; ^b^ mean and standard deviation; ^$^
*t*-student; ^&^ Mann-Whitney; ^#^ Pearson’s chi-square. *p* value in bold is significant.

**Table 2 ijerph-19-13915-t002:** Differences in baseline dietary antioxidant capacity (DaC), antioxidant capacity (aC) from food groups and coffee, Brazilian Healthy Eating Index—Revised (BHEI-R) total scores and components, and oxidative stress biomarkers between women with the occurrence of food aversions and non-occurrence of food aversion during adjuvant chemotherapy for breast cancer (n = 73).

	Occurrence of Food Aversion (n = 24)	Non-Occurrence of Food Aversion (n = 49)	*p*
DaC (mmol/d) ^a^	9.8 (7.3, 12.1)	8.4 (6.12, 12.2)	0.565 ^&^
aC from whole cereals, legumes, tubers, and roots (mmol/d) ^a^	0.79 (0.44, 155)	0.81 (0.54, 113.6)	0.721 ^&^
aC from total fruits (mmol/d) ^a^	1.21 (0.68, 1.99)	1.02 (0.51, 1.78)	0.711 ^&^
aC from cruciferous vegetables (mmol/d) ^a^	0.13 (0.04, 0.29)	0.12 (0.04, 0.24)	0.995 ^&^
aC from orange and dark green vegetables and fruits (mmol/d) ^a^	0.50 (0.28, 1.13)	0.39 (0.20, 0.75)	0.142 ^&^
aC from citric fruits (mmol/d) ^a^	0.47 (0.19, 0.98)	0.35 (0.08, 1.19)	0.762 ^&^
aC from red vegetables and fruits (mmol/d) ^a^	0.13 (0.03, 0.41)	0.17 (0.05, 0.48)	0.442 ^&^
aC from polyphenol-rich foods and beverages (mmol/d) ^a^	8.24 (5.2, 11.2)	7.25 (4.5, 11.5)	0.607 ^&^
aC from coffee (mmol/d) ^a^	7.42 (3.72, 11.2)	5.58 (3.72, 7.44)	0.967 ^&^
Total BHEI-R score	71.8 (11.2)	75.4 (8.5)	0.285 ^&^
Total fruits (0–5) ^a^	5 (3.5, 5)	5 (3.9, 5)	0.802 ^&^
Whole fruits (0–5) ^a^	5 (4.1, 5)	5 (4.9, 5)	0.691 ^&^
Total vegetables (0–5) ^a^	4.7 (2.7, 5)	5 (3, 5)	0.482 ^&^
Dark green and orange vegetables and legumes (0–5) ^a^	5 (2.8, 5)	5 (4, 5)	0.951 ^&^
Total grains (0–5) ^a^	5 (4.4, 5)	5 (5, 5)	0.175 ^&^
Whole grains (0–5) ^a^	0 (0, 0.185)	0 (0, 0.35)	0.739 ^&^
Milk and dairy products (0–10) ^b^	4.6 (0.530)	5.1 (0.434)	0.554 ^$^
Meat, eggs, and legumes (0–10) ^a^	8.7 (6.5, 10)	9.1 (7.1, 10)	0.350 ^&^
Oils (0–10) ^a^	10 (10, 10)	10 (10, 10)	0.061 ^&^
Saturated fat (0–10) ^a^	7.2 (3.5, 8.7)	6.5 (3.8, 9.4)	1.000 ^&^
Sodium (0–10) ^a^	9.0 (8.0, 9.8)	8.5 (6.9, 9.3)	0.244 ^&^
TBARS (μmol/L) ^a^	4.9 (4.3, 5.6)	4.8 (4.1, 5.9)	0.850 ^&^
HL (μmol/L) ^a^	4.3 (3.3, 5.3)	3.9 (2.9, 6.2)	0.752 ^&^
GSH (μmol/L) ^b^	76.7 (19.6)	78.8 (22.1)	0.698 ^$^
FRAP (μmol/L) ^b^	589.3 (118.1)	648.6 (164.9)	0.122 ^$^
CP (μmol/L) ^a^	0.7 (0.6, 1.1)	0.7 (0.6, 1.2)	0.504 ^&^

DaC: dietary antioxidant capacity; BHEI-R: Brazilian Healthy Eating Index Revised (BHEI-R). TBARS: thiobarbituric acid reactive substances; LH: lipid hydroperoxides; GSH: reduced glutathione; FRAP: ferric reducing antioxidant power; CP: carbonylated proteins; SoFAAS: calories from solid fat, alcohol and added sugar; ^a^ median and interquartile range; ^b^ mean and standard deviation; ^$^ *t*-student; ^&^ Mann-Whitney.

**Table 3 ijerph-19-13915-t003:** Association between dietary antioxidant capacity (DaC), diet quality according to the Brazilian Healthy Eating Index Revised (BHEI-R) and oxidative stress biomarkers and non-occurrence of food aversions (n = 73).

	Non-Occurrence of Food Aversion					
	OR (Crude)	CI95%	*p*	OR (Adjusted)	CI95%	*p*
BHEI-R score	1.04	0.987–1.096	0.137	1.080 *	1.003–1.154	**0.041**
DaC (mmol/day)	0.992	0.908–1.083	0.801	0.986 *	0.883–1.102	0.811
TBARS (μmol/L)	0.98	0.878–1.095	0.729	0.963 ^#^	0.855–1.085	0.583
LH (μmol/L)	0.992	0.831–1.184	0.932	0.924 ^$^	0.761–1.123	0.932
GSH (μmol/L)	1.005	0.981–1.029	0.693	1.002 ^+^	0.977–1.028	0.862
FRAP (μmol/L)	1.002	0.999–1.006	0.124	1.003 ^+^	0.999–1.007	0.131
CP (μmol/L)	2.192	0.560–8.580	0.259	2.540 ^$^	0.543–11.867	0.236

Reference group: women with food aversion occurrence. DaC: Dietary antioxidant capacity; BHEI-R: Brazilian Healthy Eating Index Revised (BHEI-R). TBARS: thiobarbituric acid reactive substances; LH: lipid hydroperoxides; GSH: reduced glutathione; FRAP: ferric reducing antioxidant power; CP: carbonylated proteins; OR: odds ratio; CI95%: 95% confidence interval * adjusted by age, schooling, and the number of chemotherapeutic sessions; ^#^ adjusted by age, schooling, tumor size, and alcohol intake; ^+^ adjusted by age and tumor stage; ^$^ adjusted by age and tumor size. *p* value in bold is statistically significant.

## Data Availability

The data presented in this study are available on request from the corresponding author. The data are not publicly available due to privacy.

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
