# Peer review of "Diet Quality Influences the Occurrence of Food Aversions in Women Undergoing Adjuvant Chemotherapy for Breast Cancer"

_ijerph, 2022, doi:10.3390/ijerph192113915_

Round 1

Reviewer 1 Report

Reviewer comments and suggestions

The study was a follow-up where the authors questioned about diet quality may influence the occurrence of food aversions in women going for chemotherapy for breast cancer. 

The study included 73 women with breast cancer. Women were asked if they developed food aversions during adjuvant chemotherapy. The result included that Red meat was the main aversion-causing food reported by the women (37.9%, n=9). In the adjusted regression analysis, women exhibiting higher Diet quality - Brazilian Healthy Eating Index Revised BHEI-R scores were 1.08 times more likely to not develop food aversions during adjuvant chemotherapy (p=0.041). 

Overall, the manuscript was well written. However, a few concerns/comments needed to be explained/modified. 

A few comments are below to be incorporated into the manuscript. 

  1. Line 20-21, Please mention the name of the place here as well
  2. Line 39 I think one reference is not enough to validate the line, and typo error in line 56
  3. Where was the data before therapy ( not chemotherapy) as suggested by the authors
  4. Table 2 Why it was mention double star, is they represent significant difference, seem they are not
  5. How they clarified the statements baseline data need to be presented
  6. Line 271-272 is there was the possible reason for this
  7. Line 297-299 Please discuss more on this topic
  8. All references should be modified based on the MDPI journals.

Author Response

Dear reviewer, your suggestions are very important to our manuscript. We thank you for the revisions. You will find the answers and justifications below.

1- Line 20-21, Please mention the name of the place here as well

Reply: The name of the place was included (line 20-21).

2-Line 39 I think one reference is not enough to validate the line, and typo error in line 56

Reply: This reference is from the main public/governmental cancer institute in Brazil, and the national epidemiologic information is provided only from it (INCA, 2022). We didn't find other epidemiological data on breast cancer in Brazil. You can find out at https://www.inca.gov.br/assuntos/inca

We didn't find the typo error in line 56, you can appoint so we can correct it.

3-Where was the data before therapy ( not chemotherapy) as suggested by the authors

Reply: These women were not submitted to other treatments before chemotherapy, only mammary surgery; data at T0 were collected on the diagnosis day. Then, the only data we have is shown in the tables in the results session.

4.Table 2 Why it was mention double star, is they represent significant difference, seem they are not.

5.How they clarified the statements baseline data need to be presented

6.Line 271-272 is there was the possible reason for this

Reply: We appreciate your commentary. Initially, the double star referred to the statistical test applied, not to significant difference (which is represented in bold type). We changed the representation of statistical tests, since double star is frequently used in the literature to indicate significant differences in tables 1, 2 and 3.

7.Line 297-299 Please discuss more on this topic:

Reply: We appreciate your suggestion, it was important to clarify the possible explanations for the results. We included a discussion of this topic, highlighted in yellow - lines 318-320.

8.All references should be modified based on the MDPI journals.

Reply: Modified for MDPI journal style.

Reviewer 2 Report

The authors take up an interesting and important topic of manuscript. Abstract. It needs some refinement. The authors do not write here what the novelty under development is. The Line 24 authors write that: "Red meat was the main aversion-causing food reported by the women (37.9%, n = 9)." This is quite a chaotic sentence because we do not know what patients have to choose from when it comes to diet. We don't know if they all ate red meat or if there were vegans or vegetarians among them. The inline sentences 28–30 add little to the value of the job. I recommend that you provide more substantive conclusions. I recommend that the authors describe the studied women in more detail, especially in terms of the diet they had, age, severity of the disease, or duration of the disease. The work lacks information on how the research was carried out with the use of which method. The point concerning the methodology requires a solid refinement from the substantive point of view. The description should definitely be detailed. This point needs some fine-tuning. The authors briefly mention the applied statistical methods without providing formulas or explanations. The strength of the research is the original and very interesting topic taken up by the authors and the methodology used, while the weakness of the work is the conclusion. I recommend that the authors write a conclusion in relation to the purpose of the work, applied methodology, and achieved goals. The literature review is solidly conducted on the basis of the latest studies, which is of value for the work. 

Author Response

REVIEWER 2:

Dear reviewer, your suggestions are very important to our manuscript. We thank you for the revisions. You will find the answers and justifications below.

1.The authors take up an interesting and important topic of manuscript. Abstract. It needs some refinement. The authors do not write here what the novelty under development is.

Reply: We appreciate your suggestion. In fact, the novelty of the research helps to add value to it. We included the novelty in the conclusions section of the abstract, as you suggested (lines 28-32).

2.The Line 24 authors write that: "Red meat was the main aversion-causing food reported by the women (37.9%, n = 9)." This is quite a chaotic sentence because we do not know what patients have to choose from when it comes to diet. We don't know if they all ate red meat or if there were vegans or vegetarians among them.

Reply: Your statement remarks are highly relevant and significant. Herein, “food aversions” refers to aversion to food developed in response to adjuvant chemotherapy. In other words, foods that were previously consumed as part of a habitual diet, and due to the adverse effect of treatment, developed an aversion to it. Please, you can see additional details about the collected data in the Methodology section, “2.5. Food Aversions Assessment"- lines 146-155.

3.The inline sentences 28–30 add little to the value of the job. I recommend that you provide more substantive conclusions.

Reply: We appreciate your suggestion since the conclusion section is one of the most important to add value to the research. We improved the abstract conclusions as you suggested  (lines 28-32).

4.I recommend that the authors describe the studied women in more detail, especially in terms of the diet they had, age, severity of the disease, or duration of the disease. The work lacks information on how the research was carried out with the use of which method. The point concerning the methodology requires a solid refinement from the substantive point of view. The description should definitely be detailed. This point needs some fine-tuning.

Reply: The sample feature is very important for the interpretations of results from the research, and we appreciate your concern. Data on food intake was collected by the application of a validated food frequency questionnaire, before the beginning of adjuvant chemotherapy, and this method was used to calculate the Brazilian Healthy Eating Index Revised (BHEI-R), which is a method that reflects the adequation of the habitual diet of these women based on the dietary recommendations for the Brazilian population. Furthermore, we included a detailed explanation on dietary habits of these women in lines 219-222. Table 1 shows the classification of the tumor stage (highlighted in yellow), and this information reflects the severity of the disease; in addition, tumor size is related to the aggressiveness of the disease, revealing its severity (ZHANG et al., 2021; ACS, 2022). We included other result on severity of disease in lines 215-217 (highlighted in yellow). The mean age of the sample is described in line 207, in result section (51.9±11.6 years), and by occurrence (50.6±10.2 years) and non-occurrence of food aversion (52.5±12.3 years) in Table 1 (highlighted in yellow). The data on the duration of the disease is not available in our database, since this is a prospective study with two punctual periods of data collection; the food aversion assessment was collected at the end of adjuvant chemotherapy cycles, but we had no access to information of heal from the disease for these participants, since the end of chemotherapy cycles is not necessarily the end of all treatment. In the result section, lines 210-211 (highlighted in yellow), you will find the average chemotherapy duration for the present sample (14.1±4.4 months).

Nevertheless, description of the studied women was improved in the Results section.

5.The authors briefly mention the applied statistical methods without providing formulas or explanations.

We appreciate your concern and description of statistical analyses was improved in the manuscript (lines 185-205).  

6.The strength of the research is the original and very interesting topic taken up by the authors and the methodology used, while the weakness of the work is the conclusion. I recommend that the authors write a conclusion in relation to the purpose of the work, applied methodology, and achieved goals. The literature review is solidly conducted on the basis of the latest studies, which is of value for the work.

Reply: We appreciate your concern about the strength and weaknesses of the work and its value. The conclusion (lines 331-332, highlighted in yellow) shows the result found in relation to the purpose of it and achieved goals: “This investigation showed that diet quality before adjuvant chemotherapy may influence the non-occurrence of food aversion..”, and the strength of it, such as innovative research and practical application of it.  

REFERENCES

Instituto Nacional de Câncer José Alencar Gomes da Silva. Estimativa 2020: incidência de câncer no Brasil. 2020a. https://www.inca.gov.br/sites/ufu.sti.inca.local/files/media/document/estimativa-2020- incidencia-de-cancer-no-brasil.pdf. Accessed 10 Jul 2022

Zhang F, de Haan-Du J, Sidorenkov G, Landman GWD, Jalving M, et al. Type 2 Diabetes Mellitus and Clinicopathological Tumor Characteristics in Women Diagnosed with Breast Cancer: A Systematic Review and Meta-Analysis. Cancers (Basel). 2021;13(19):4992. doi: 10.3390/cancers13194992.

American Cancer Society. Breast cancer early detection and diagnosis.  https://www.cancer.org/cancer/breast-cancer/screening-tests-and-early-detection/breast-biopsy.html. Accessed 06 de aug 2022.

Reviewer 3 Report

Authors have performed an study to evaluate food aversion in women undergoing adjuvant chemotherapy for breast cancer. In my opinion, this is a really nice study greatly performed that includes in the introduction all the necessary background to understand why this study is done; the research design is adequate and methods are clearly described. 

Results and discussion does clearly expose the results obtained and contrast them with recent references, and conclusion answer all the objectives mentioned on the introduction. 

I would only reccomend the authors to modify in the introduction the first paragraph, as they expose "(...) new diagnoses of breast cancer are expected for each year of the 2020-2022 trienium" and this does not make sense when we are in October 2022.  

Author Response

REVIEWER 3:

Dear reviewer, your suggestions are very important to our manuscript. We thank you for the revisions. You will find the answers and justifications below.

Authors have performed an study to evaluate food aversion in women undergoing adjuvant chemotherapy for breast cancer. In my opinion, this is a really nice study greatly performed that includes in the introduction all the necessary background to understand why this study is done; the research design is adequate and methods are clearly described.

Results and discussion does clearly expose the results obtained and contrast them with recent references, and conclusion answer all the objectives mentioned on the introduction.

1.I would only recommend the authors to modify in the introduction the first paragraph, as they expose "(...) new diagnoses of breast cancer are expected for each year of the 2020-2022 trienium" and this does not make sense when we are in October 2022.

Reply: We appreciate your concern about the updated information. Since the year 2022 is not over, we have no more updated epidemiological data on breast cancer worldwide, described in scientific literature. In addition, this is the last published information on the global epidemiology of cancer, provided by the Global Cancer Observatory (GLOBOCAN) inserted in the International Agency for Research on Cancer (IARC, World Health Organization). You can find out in https://gco.iarc.fr/

Round 2

Reviewer 2 Report

The authors improved the manuscript in line with the suggestions and recommendations. The article looks very good. Thank you for the effort.